# Rotating waves during human sleep spindles organize global patterns of activity that repeat precisely through the night

Lyle Muller[1], Giovanni Piantoni[2], Dominik Koller[1], Sydney S Cash[2], Eric Halgren[3,4], Terrence J Sejnowski[1]*

[1]Computational Neurobiology Laboratory, Salk Institute for Biological Studies, La Jolla, United States; [2]Department of Neurology, Massachusetts General Hospital, Harvard Medical School, Boston, United States; [3]Department of Radiology, University of California, San Diego, San Diego, United States; [4]Department of Neurosciences, University of California, San Diego, San Diego, United States

**Abstract** During sleep, the thalamus generates a characteristic pattern of transient, 11-15 Hz sleep spindle oscillations, which synchronize the cortex through large-scale thalamocortical loops. Spindles have been increasingly demonstrated to be critical for sleep-dependent consolidation of memory, but the specific neural mechanism for this process remains unclear. We show here that cortical spindles are spatiotemporally organized into circular wave-like patterns, organizing neuronal activity over tens of milliseconds, within the timescale for storing memories in large-scale networks across the cortex via spike-time dependent plasticity. These circular patterns repeat over hours of sleep with millisecond temporal precision, allowing reinforcement of the activity patterns through hundreds of reverberations. These results provide a novel mechanistic account for how global sleep oscillations and synaptic plasticity could strengthen networks distributed across the cortex to store coherent and integrated memories.

*For correspondence: terry@salk.edu

## Introduction

Memories are stored in distributed networks across the cortex. In the two-stage model of memory consolidation (*McClelland et al., 1995*; *Rasch and Born, 2007*), memories are integrated in the hippocampus and then linked in the neocortex for long-term storage, where information represented in visual, auditory, somatosensory, or cognitive regions must be bound into a coherent whole (*Wheeler et al., 2000*; *Horner et al., 2015*). It is well established that sleep oscillations actively contribute to this process: during stage 2 sleep spindles, the thalamus generates a rhythmic activity pattern that becomes widespread through large-scale thalamocortical loops (*Contreras et al., 1996*), and spindles are critical to sleep-dependent memory consolidation (*Gais et al., 2002*; *Mednick et al., 2013*). Long-range connections in cortex result primarily from excitatory pyramidal cells (*Sholl, 1956*; *Schüz et al., 2002*), but precisely how sleep oscillations aid strengthening of these excitatory connections between distributed cortical networks through spike-time dependent plasticity (STDP) remains unclear, particularly in the presence of long axonal conduction delays (*Lubenov and Siapas, 2008*). Here, we identify a global activity pattern repeatedly observed during sleep spindle oscillations in human neocortex that could serve this role.

We study intracranial electrocorticogram (ECoG) recordings of five clinical patients in stage 2 sleep and apply recently developed computational methods (*Muller et al., 2014*) to classify

**eLife digest** When you wake up in the morning after a good night's sleep you feel refreshed. You can also think more clearly because your memory has been re-organized, a process called memory consolidation. The problem that the brain has to solve during sleep is how to integrate memories of experiences that happened during the day with old memories, without losing the older memories.

Scientists know that waves of electrical activity, referred to as spindles, help to consolidate and integrate memories during sleep. Spindles are active in the cerebral cortex, the part of your brain used for thinking, in the time between dream sleep and deep sleep. Yet it is not known exactly how these bursting patterns of electrical activity help to strengthen memories.

Now, Muller et al. explored how the spindles could strengthen and connect parts of memories stored in distant parts of the brain. First, a computer algorithm analyzed electrical recordings of brain activity taken while five patients with epilepsy slept. The patients were being monitored to help with their seizures, and the recordings showed that spindles do not occur at the same time throughout the cortex as previously thought. Instead, the spindle is a wave that begins in portion of the cortex near the ear, spirals through the cortex toward the top of back of the head and then on to the forehead area before circling back.

These repeated circular waves of electrical activity strengthen connections between brain cells in distant parts of the brain. For example, these waves may help strengthen connections between the cells of the cortex that separately store memories of the sound, sight and feel of an event during the day, whether that's being bitten by a dog or talking with a friend. Next, Muller et al. plan to develop computer models of the spindles and verify whether their models make accurate predictions by studying spindles in sleeping mice and rats.

spatiotemporal dynamics at the level of individual oscillation cycles. ECoG arrays were implanted in subjects undergoing evaluation for resective surgery of epileptogenic cortex (*Figure 1A*, left). Over several days of recording, these subjects exhibit long periods without major epileptic events. During that time, subjects express a relatively normal sleep architecture, with well-defined sleep oscillations. Stage 2 sleep epochs were then manually identified by an expert rater, and sleep spindles recorded on the ECoG were isolated using automated techniques (*Hagler et al., 2016*). These spindles appeared physiologically normal and well-isolated from background noise (*Figure 1A*, right and *Figure 1—figure supplement 1*). Our algorithmic approach classifies spatiotemporal patterns as expanding waves, defined as a significant linear increase in phase offset with distance from a point source (*Figure 1—figure supplement 2*; see Materials and methods – Spatiotemporal dynamics), or rotating waves, defined as a significant increase in phase offset with rotation about a wave center (*Figure 1—figure supplement 3*). In 41,860 spindle oscillation cycles tested across subjects, a large proportion (50.8%) was classified as rotating waves, along with a smaller subset (15.6%) as expanding. After inspecting these results, we observed further that the rotating waves exhibited a clear bias towards travel in the temporal → parietal → frontal (TPF) direction (69.5%, $p<10^{-10}$, one-tail binomial test against equal occurrence, 14,796 TPF cycles, 21,272 total; for each individual subject $p<10^{-3}$, see *Figure 1—source data 1* for individual wave totals) (*Figure 1B* and *Video 1*). Propagation speed distributions peaked between 2–5 m/s (*Figure 1C*), varying within a narrow range from the 20th to the 80th percentiles (3–9 m/s for the full distribution; 4–10, 3–8, 3–10, 2–3, and 4–13 m/s for individual subjects, respectively), within the range of conduction speeds for the short (*Girard et al., 2001*) and long (*Schüz et al., 2002*; *Swadlow and Waxman, 2012*) white matter association fibers. Further, this rotating TPF organization occurred consistently across subjects and implantation hemispheres (*Figure 1D*; see also *Figure 1—figure supplements 5–7*).

Spike-time dependent plasticity is a well-studied mechanism for regulating synaptic strengths that depends on the relative timing of presynaptic inputs and postsynaptic spikes (*Markram et al., 1997*; *Bi and Poo, 1998*), but for establishing large-scale neural assemblies during sleep oscillations through synaptic plasticity, axonal conduction delays pose a specific problem (*Lubenov and Siapas, 2008*). For example, cortical white matter association fibers have conduction delays up to 50

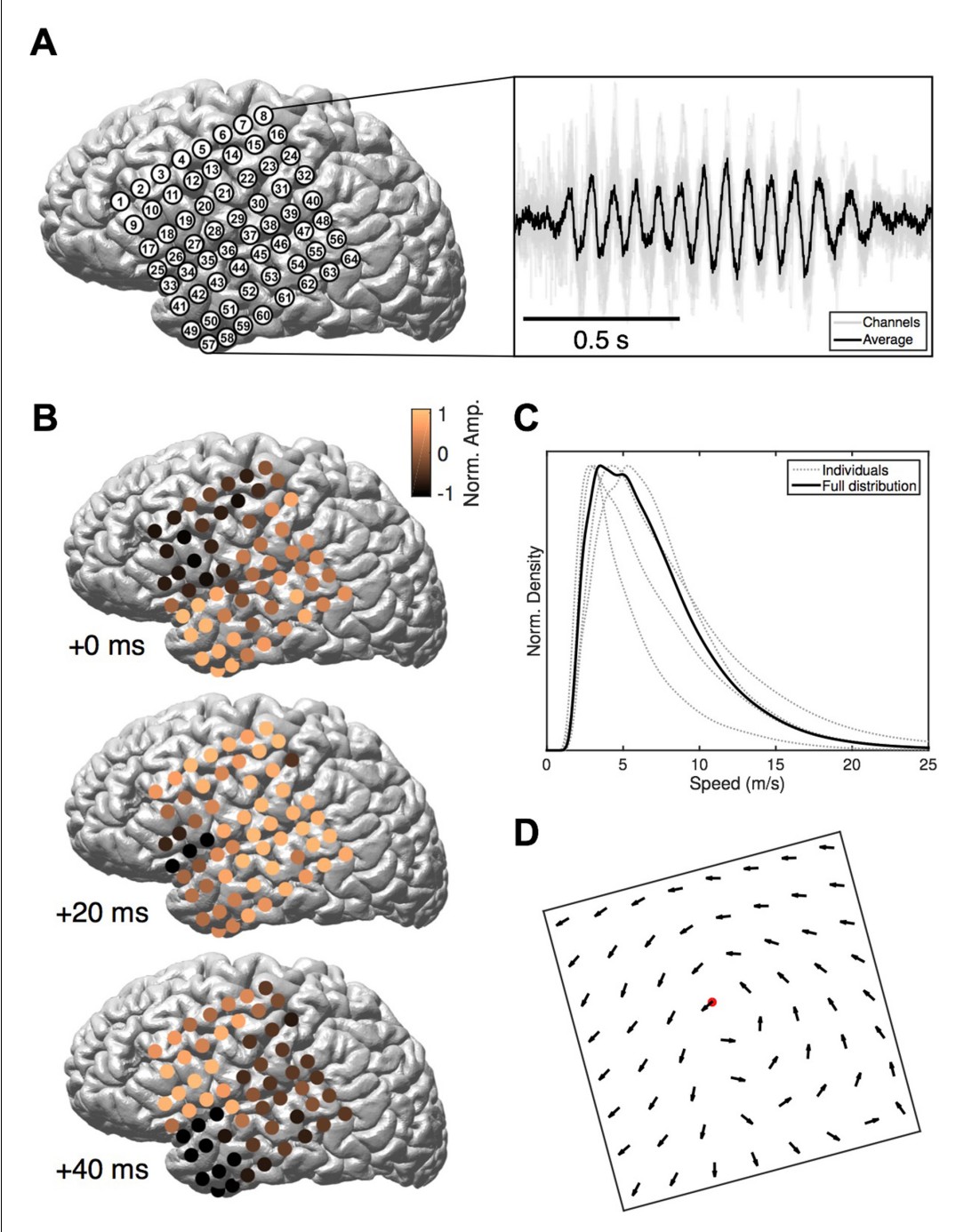

**Figure 1.** Rotating waves during spindles. (A) Electrode placement for subject 1 (left), with a stereotypical spindling epoch observed on the array (right). The right panel depicts the average over channels (black) together with the individual channels (gray). (B) When visualized on the cortex, individual spindle cycles are often organized as rotating waves traveling from temporal (+0 ms, top) to parietal (+20 ms, middle) to frontal (+40 ms, bottom) lobes. (C) Phase speed distributions across subjects. Plotted is the kernel smoothing density estimate for individual subjects (gray dotted lines) and for the full distribution (black line). (D) The field of propagation directions, aligned on the putative rotation center and averaged across oscillation cycles and across subjects, shows a consistent flow in the temporal → parietal → frontal (TPF) direction. The center point is marked in red.

The following source data and figure supplements are available for figure 1:

**Source data 1.** Patient information and wave classification totals.

**Figure supplement 1.** Power spectral density and spatial correlation analysis.

*Figure 1 continued on next page*

*Figure 1 continued*

**Figure supplement 2.** Method for isolating expanding spatiotemporal patterns.

**Figure supplement 3.** Method for isolating rotating spatiotemporal patterns.

**Figure supplement 4.** Analysis of 1528 spindles in Subject 1.

**Figure supplement 5.** Vector field averaging controls.

**Figure supplement 6.** Vector field distribution control.

**Figure supplement 7.** Vector field averages for each subject.

**Figure supplement 8.** Distribution of rotation center.

**Figure supplement 9.** Summary statistics across subjects.

**Figure supplement 10.** Consistent, coherent phase flow during spindles occurs uniquely in the 9–18 Hz frequency band.

**Figure supplement 11.** Robustness to noise and center position.

**Figure supplement 12.** Local versus global simulated rotating waves.

milliseconds across the cortex (*Figure 2A*, left) (*Girard et al., 2001*; *Schüz et al., 2002*; *Swadlow and Waxman, 2012*).

It is well established that spindles cause pyramidal cells and interneurons in cortex to fire preferentially at the peak of the surface-positive (depth-negative) LFP oscillation, both in intracellular (*Contreras and Steriade, 1995*, *1996*; *Kandel and Buzsáki, 1997*) and extracellular (*Peyrache et al., 2011*) recordings. If cortical spindles were perfectly synchronized, spikes emitted during one cycle of the spindle oscillation would arrive at their post-synaptic targets with this temporal delay, leading to a pairing within the window for persistent long-term depression (LTD) that would progressively weaken long-range connections (*Figure 2A*, right). If, however, spindles are self-organized into large-scale wave-like activity patterns, with phase speeds matching those of the underlying fiber networks and stereotyped, precisely repeating trajectories (*Figure 2B*, left), then EPSPs caused by spikes traveling along pyramidal axons to distant regions in the cortex would align with the local burst of population activity (*Figure 2B*, right), creating the conditions necessary for synaptic strengthening to occur.

Next, we wanted to understand whether these population activity patterns repeat with the temporal precision required for strengthening of

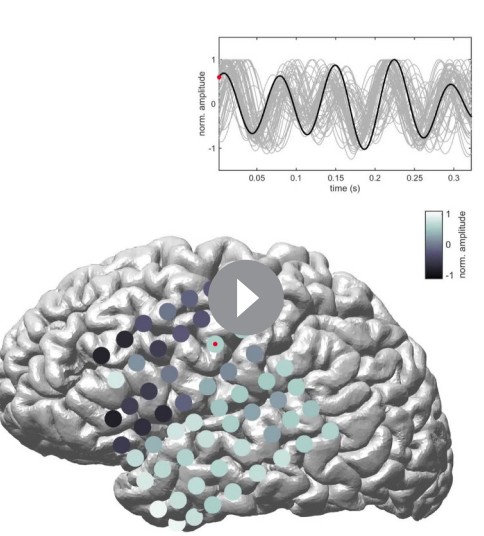

**Video 1.** Rotating waves over five spindle oscillation cycles. Normalized activity for bandpass filtered timeseries is plotted in falsecolor at electrode positions on the cortical surface of Subject 1. The cortical electrode marked with a red dot (bottom) corresponds to the black timecourse in the inset (top). The other ECoG channels are plotted in gray. The time period visualized corresponds to approximately 300 milliseconds, or five cycles of the spindle oscillation. Note that no spatial smoothing is applied in these data.

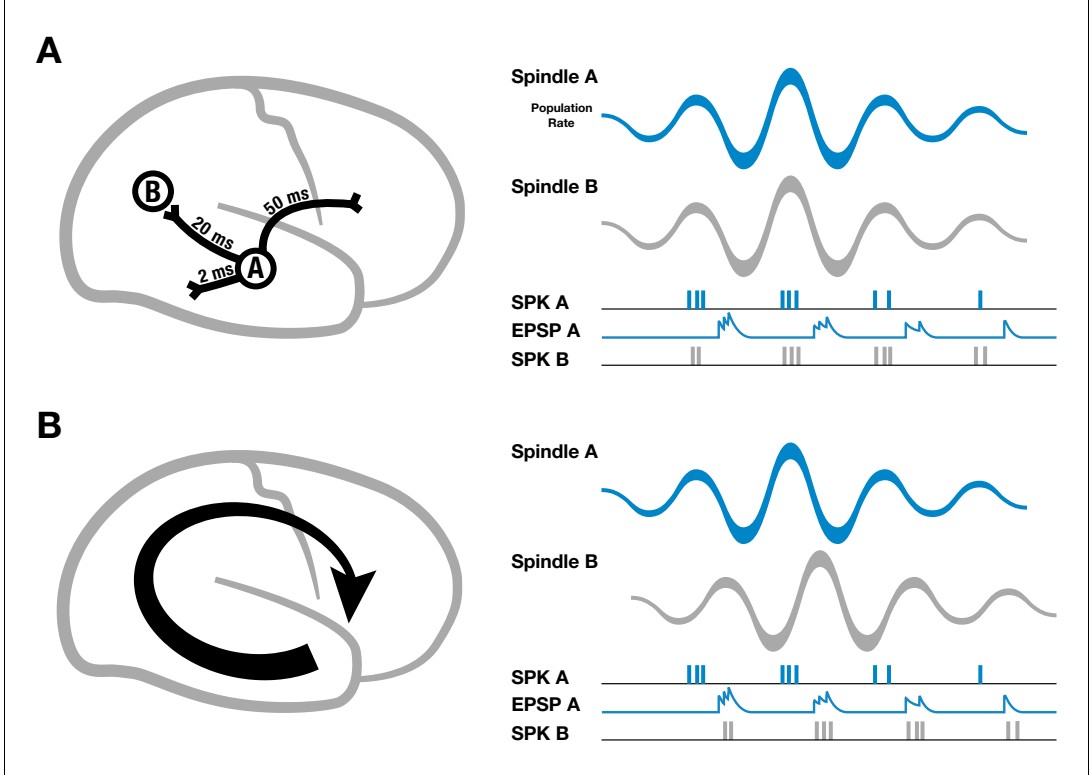

**Figure 2.** Schematic of spindles and axonal delays. (A) Spikes emitted from region A will arrive at B with a temporal delay of 20 milliseconds (left). If spindle oscillations were perfectly synchronized across the cortex, EPSPs from region A would occur after the spikes in region B, within the window for long-term depression (right). (B) In contrast, if spindles are spatiotemporally organized with stereotyped trajectories (left), then EPSPs from region A would align with population spiking in region B, allowing for synaptic strengthening to occur.

large-scale assemblies. We defined the correlation magnitude over phase values on the electrode array between individual oscillation cycles to be a pairwise similarity index (see Materials and methods), in order to detect similar spatiotemporal patterns across oscillation cycles (*Video 2*). By calculating this metric over all cycle pairs in different wave classes (all cycles, expanding, rotational), we can directly compare the temporal precision mediated by each type. The cumulative distribution function (CDF) of similarity indices among identified rotational waves is highly shifted to the right (*Figure 3A*, black) compared to the CDF for all cycles (5 subjects, 42/54 sleep epochs, 77.8% significant, one-tailed two-sample Kolmogorov-Smirnov test, $\alpha = 0.01$, Bonferroni correction), indicating higher intra-class similarity between these cycles than for other wave types. Note that this is not simply a consequence of the rotational phase pattern itself, as expanding waves emanating from a consistent point source could certainly exhibit higher intra-class similarity than rotational waves with a varying center. Further, the median similarity index consistently increases in individual subjects when rotational waves of progressively increasing strength are considered (*Figure 3B*). This

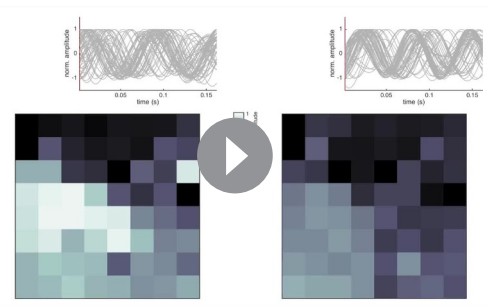

**Video 2.** Rotating waves with high spatiotemporal similarity. Two rotating waves with high phase similarity on the ECoG array, separated by 5.62 min of stage 2 sleep. Bandpass filtered timeseries are normalized to their maximum within the interval and plotted in falsecolor (bottom panels). Activity for each channel is plotted as a function of time (top panels), with an indication of temporal progression (red dotted line).

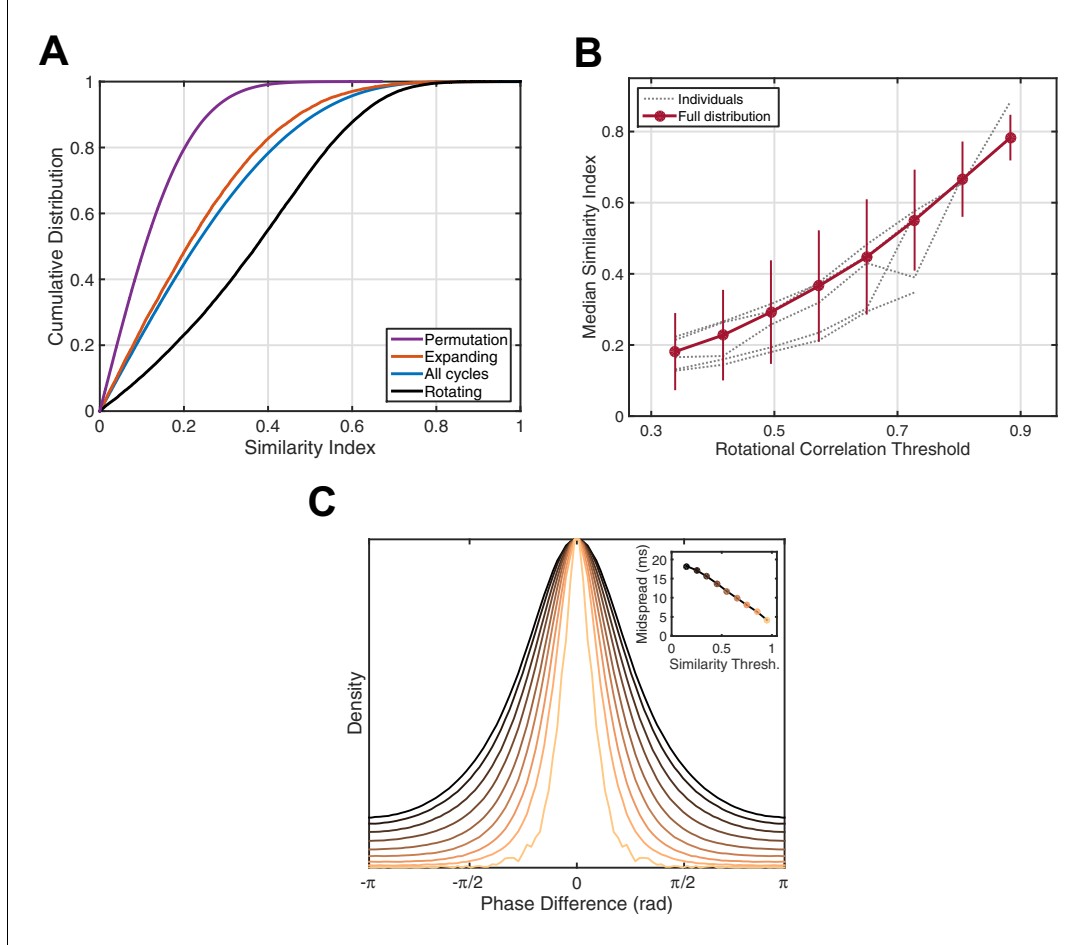

**Figure 3.** Phase pattern analysis. (**A**) Rotating waves exhibit higher intra-class similarity. Cumulative distribution functions (CDFs) for shuffled data (purple), expanding waves (red), all cycles (blue), and rotating waves (black) are given for an example 15 min epoch of stage 2 sleep (subject 5). (**B**) Spindle cycles exhibiting stronger rotating patterns also express greater intra-class similarity. Gray lines indicate the median similarity index (ordinate) for the population of oscillation cycles expressing rotational waves above a threshold strength (abscissa), averaged over individual sleep epochs. Red dots and error bars indicate the median and median absolute deviation for the full distribution, respectively. (**C**) Spindle cycles exhibiting high similarity index are temporally precise. The distribution of phase difference at each electrode across spindle cycles is given as a function of the similarity index (indicated by colors, inset). By utilizing the mean spindle oscillation frequency (13.5 Hz), the midspread (interquartile range) of each distribution is given in units of time (inset).

indicates the observed rotating waves strongly modulate temporal precision in repeated patterns of population activity. To be specific, by utilizing the average temporal frequency for these spindle oscillations (13.5 Hz), we can estimate that in two cycles whose similarity index falls into the highest bin in *Figure 3C* (0.9–1.0), 50% of electrodes will experience an alignment of the spatiotemporal activity pattern within a 5 millisecond temporal window. Recent experiments have shown a tight temporal link between field potentials and synaptic currents (both EPSPs and IPSPs; *Haider et al., 2016*). By detecting these precisely recurring activity patterns in ECoG recordings, we can infer that distributed networks composed of local excitatory and inhibitory groups, whose firing is modulated by thalamocortical fibers during the sleep spindle, are repeatedly activated with a millisecond accuracy that is well within the temporal precision required for STDP.

If this millisecond precision in fact mediates formation and maintenance of corticocortical assemblies, we would then expect spiking associated with these synaptic currents to drive increased reverberation throughout the night, as excitatory connections between local groups of pyramidal cells and interneurons are strengthened and in turn promote more replay of the expressed activity pattern. High gamma-band power (HGP, 80–120 Hz), a reliable electrophysiological correlate of spiking

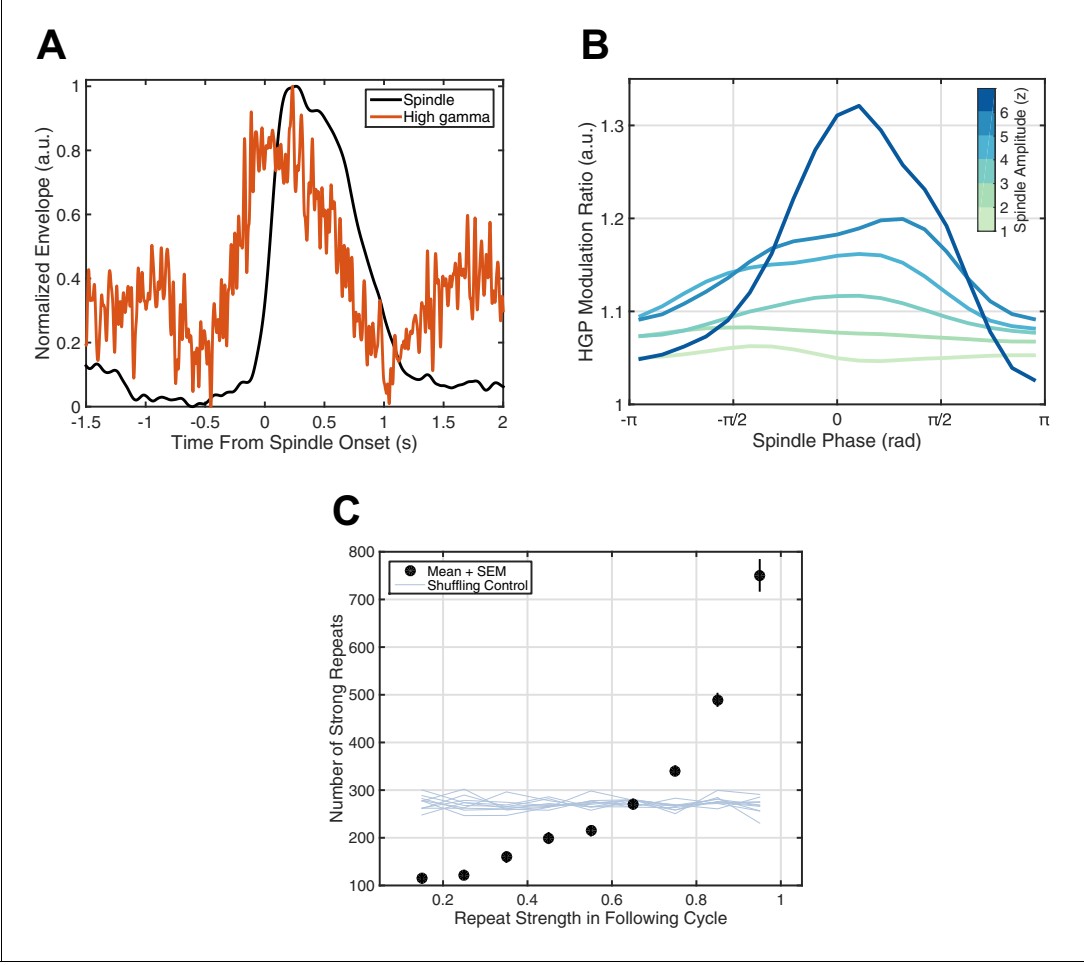

**Figure 4.** Spiking activity and increased reverberation. (**A**) High gamma-band power (HGP) consistently increases around spindle onset. Plotted are the normalized amplitude envelopes for spindles (black) and HGP (red), averaged over 186 spindles in Subject 1. (**B**) HGP is modulated by spindle phase. Plotted is the mean high gamma-band power at each phase of the spindle oscillation (20 bins), for varying amplitudes of the spindle oscillation (see colorbar), each normalized by the mean HGP in matched non-spindle epochs. (**C**) Strength of the first repeat predicts the number of strong reverberations. The number of similar rotational patterns (above similarity index 0.7) following a spindle oscillation cycle is given as a function of the next cycle's similarity index (black dots, mean + SEM) over 2.5 hr of sleep in Subject 1. Error bars are obscured by markers. Light blue lines indicate results from a shuffling permutation test (10 iterations).

The following figure supplements are available for figure 4:

**Figure supplement 1.** Strong increase of reverberation observed for rotating, but not expanding, waves.

**Figure supplement 2.** Increase in reverberation observed across subjects.

**Figure supplement 3.** Modulation of high-gamma power (HGP) by spindle phase across subjects.

activity (*Ray et al., 2008*; *Ray and Maunsell, 2011*; *Ray, 2015*; similar in nature to the 'broadband power shift' described in *Manning et al., 2009*), consistently increases around spindles (*Figure 4A*). Further, HGP is modulated by spindle phase (*Figure 4B*), increasing towards the surface-positive (depth-negative) peak, consistent with previous animal (*Peyrache et al., 2011*) and human (*Andrillon et al., 2011*) recordings. Finally, by studying repeats of rotating waves over 2.5 hr of continuous sleep recording in Subject 1, we observe a preliminary indication of increased reverberation consistent with strengthening of distributed excitatory networks: similarity in the next identified rotating wave is highly predictive of the number of strong reverberations throughout the night (black

dots, *Figure 4C*). Randomizing the relationship between the next rotating wave and the rest of the sleep recording eliminates this effect (shuffling control, *Figure 4C*), and such increased reverberation is not observed for expanding waves under similar conditions (*Figure 4—figure supplement 1*). Similar observations are consistent across subjects (*Figure 4—figure supplement 2*). These results support the hypothesis that precisely repeating rotating waves may enable strengthening of large-scale corticocortical assemblies throughout the night.

Early animal sleep spindle studies, using up to 8 electrodes in a linear array (*Andersen et al., 1967*; *Kim et al., 1995*; *Contreras et al., 1996*, *1997*), in addition to preliminary EEG evidence in the human (*Achermann and Borbély, 1998*), proposed that spindles involve global synchronization of cortical circuits, raising the possibility that this sleep oscillation places neocortex into a specialized state for consolidation of long-term memories. In recent years, several studies have reported a mixture of 'local' and 'global' spindles using amplitude-duration thresholding approaches (*Nir et al., 2011*; *Andrillon et al., 2011*). By carefully studying the phase information in the spindle frequency band recorded on large-scale ECoG arrays, we have uncovered that a substantial number of spindle oscillation cycles are organized into global, hemisphere-spanning patterns of rotating and expanding waves (*Figure 1—figure supplement 7*). These patterns most likely represent the characteristic spatiotemporal organization of the 'global' spindles observed in Andrillon et al. (*Andrillon et al., 2011*) ($\geq$40% involvement, cf. their Figure 5C), with more localized patterns left unclassified by our detection approach (*Figure 1—figure supplements 9* and *12*). These global patterns are likely established through widespread thalamocortical loops, placing the cortex into a state of large-scale coherence (*Contreras et al., 1996*), shaped into rotating and expanding waves through corticocortical white matter connections with axonal conduction speeds consistent with the observed propagation speeds (*Figure 1C*). Future computational modeling work will address in detail the role of thalamocortical, corticocortical, and corticothalamic connections in generating the spatiotemporal activity patterns reported here.

Spindles have recently been specifically and causally implicated in the sleep-dependent consolidation of long-term memories (*Mednick et al., 2013*; *Hennies et al., 2016*). While some memories integrate content from single sensory modalities, requiring consolidation in only single cortical regions (such as motor cortex, *Khazipov et al., 2004*), many memories integrate multimodal sensory and cognitive information (*Gibson and Maunsell, 1997*), and require 'global' integration of distributed networks across the cortex. In this work, we have identified a novel mechanism by which this process could occur: the stereotyped activity patterns reported here may enable STDP to establish large-scale neuronal assemblies at scales where axonal conduction delays are long relative to the oscillation cycle (*Fries, 2005*; *Lubenov and Siapas, 2008*), and repeat many times throughout sleep with millisecond accuracy. While the schema illustrated in *Figure 2* is a highly simplified view of the microscale interactions between long-range excitatory projections and local networks during spindle oscillations, computational and theoretical studies have previously obtained a detailed understanding of STDP dynamics with neurons receiving sequenced (*Rao and Sejnowski, 2001*, *2003*), bursting (*Song et al., 2000*), and oscillating inputs (*Muller et al., 2011*; *Luz and Shamir, 2016*). This theoretical understanding of the interplay between STDP and population activity can allow in future work a precise account of how microscale synaptic interactions are shaped by global oscillation patterns, and how variability in these patterns (e.g. variation in wave speed, *Figure 1C*) will affect this mechanism. Taken together, these results provide insight into how distributed information stored across cortical regions may be bound into a coherent, integrated, but specific memory through spike-time dependent synaptic plasticity.

## Materials and methods

### Subjects

Patients with longstanding pharmacologically resistant complex seizures gave fully informed consent according to NIH guidelines as monitored by the local Institutional Review Board (Massachusetts General Hospital). Electrocorticogram (ECoG) recordings during natural sleep were made over the course of clinical monitoring for spontaneous seizures. Electrode placement was determined solely by clinical criteria, with electrode grids usually spanning the Sylvian fissure and multiple lobes of the cerebral cortex (frontal, parietal, and temporal). Patients were informed that participation in the

research would not alter their clinical treatment in any way, and that they may withdraw their consent at any time without jeopardizing clinical care.

## Electrodes

ECoG contacts (Ad-Tech Medical Instrument Corp., Racine, WI) were 3 mm platinum-iridium (90% platinum) discs arranged in a two-dimensional grid (8 rows and 8 columns, Subjects 1, 3, and 4; 8 rows and 12 columns, Subject 2; 8 rows and 6 columns, Subject 5) implanted semi-chronically on the pial surface in an effort to localize the seizure origin. Within the grid, electrodes were spaced 10 mm apart. In some patients, linear ECoG arrays provided additional spatial coverage; application of our multichannel detection approach, however, focused on the two-dimensional electrode grid. One strip of electrodes positioned over the pial surface and facing the skull served as the reference during the recordings; results were additionally verified using an average reference. Note that due to reference artifacts, an average reference was employed for the recordings in Subject 4. We note as well that the temporal extent of the waves, over tens of milliseconds, makes electrophysiological artifacts such as volume conduction an unlikely explanation for the observations reported here. Recordings were performed with clinical EEG monitoring equipment (XLTEK, Natus Medical Inc., Pleasanton, CA) and sampled at 500 or 512 Hz.

## Electrode localization

Post-implantation electrode localization utilized coregistration of preoperative magnetic resonance imaging (MRI) with postoperative computed tomography (CT), as described by Dykstra et al. (*Dykstra et al., 2012*). Cortical surfaces were computed with FreeSurfer (*Dale et al., 1999*; *Fischl et al., 1999*). To account for the misalignment between the MRI and CT due to the craniotomy, the locations of the grid electrodes were projected onto the cortical surface (*Dykstra et al., 2012*). Geodesic electrode distances, which take into account the folded geometry of the cortical surface and were used in some calculations (e.g. estimation of spatial correlation values), were estimated using a shortest paths approach on the cortical surface mesh.

## Sleep spindle detection

During the monitoring period for spontaneous seizures, the subjects slept in the clinical environment and expressed relatively normal sleep patterns. ECoG recordings that did not have a seizure in the preceding or following 12 hr were scored visually by an expert rater following the standard sleep stage classification (*Silber et al., 2007*). For each patient, we obtained from 15 to 101.5 min of NREM stage 2 sleep, when the spindles are most prevalent. Individual sleep spindles were then detected during stage 2 sleep using one of several complementary methods, either based on amplitude-duration thresholding (*Gais et al., 2002*; *Warby et al., 2014*) or a similar wavelet-based approach with additional verification steps (*Hagler et al., 2016*). The number of spindles detected was in agreement with previous reports of spindle density (*Gais et al., 2002*; *Warby et al., 2014*).

The results were additionally verified using a novel approach quantifying the signal-to-noise ratio (SNR) of power in the bandpass (9–18 Hz, 8th-order Butterworth filter) versus the bandstop (1–100 Hz bandpass, with 9–18 Hz bandstop) signal. In this approach, the SNR metric is calculated on short (500 ms) sliding windows in each channel. When the SNR metric reaches 0 dB, the signal and noise power are at parity, corresponding to a sharp, narrowband epoch in the recording. Picking a constant SNR threshold (5 dB) corresponds roughly to the constant false alarm rate (CFAR) technique in radar. This approach yields a conservative but approximately amplitude-invariant method for detecting arbitrary narrowband epochs in multichannel data.

## Power spectral density analysis

Following spindle detection, we made a verification analysis by calculating the average power spectral density (PSD) over isolated spindle epochs in each subject. Data were initially filtered to remove line noise artifacts, and PSDs were then calculated in 1 s intervals during the spindle and matched non-spindle epochs. PSDs for individual channel and spindle epochs were concatenated into a large array and averaged in each case. A clear peak in the 11–15 Hz frequency band for the spindle epochs can be seen, while no peak is observed in the matched non-spindle epochs (*Figure 1—figure supplement 1A*). Divisive normalization is calculated by dividing the power at each frequency in

the spindle epochs by the power in the matched non-spindle epochs, and expressing the result in dB (*Figure 1—figure supplement 1A*, inset). If the divisive normalization over an epoch of stage 2 sleep reached 5 dB, then the spindles were taken to be well-isolated and possessing the spectral characteristics necessary for an accurate phase representation, and were then included in further analysis. Based on this calculation, 54 individual epochs of stage 2 sleep, varying from 30 s to 35 min in duration, were selected in five clinical subjects.

## Temporal filtering

Temporal filtering of stage 2 sleep recordings was carried out with an 8th-order digital Butterworth bandpass filter (9–18 Hz), forward-reverse in time to prevent phase distortion (see MATLAB function filtfilt). All results were checked with multiple cutoff frequencies to ensure against parametric sensitivity.

## Spatial correlation analysis

To assess spatial correlation during spindle oscillations as a function of distance in the cortex, we adapted standard methods (*Destexhe et al., 1999*) with a Monte Carlo implementation more suited for sampling correlations on two-dimensional electrode arrays. To calculate this metric, one electrode is first selected at random, and a second is then selected from the set of electrodes within a binned distance $d_i$ from the first. The temporal correlation between these electrode pairs is then computed in the bandpass timeseries between the start and end points of the spindle. This process is repeated for a given number of iterations $N_k$ at each distance bin $d_i$, and the average spatial correlation is computed as the mean of the correlation values for the individual epochs (*Figure 1—figure supplement 1B*, black). The spatial correlation values were computed for non-spindle epochs matched to the temporal extent of the individual tested spindle epochs (*Figure 1—figure supplement 1B*, red). The average spatial correlation values are elevated during spindle oscillations with respect to the matched non-spindle periods of stage 2 sleep, indicating that a coherent, large-scale increase in global activity occurs during spindles, in agreement with previous studies (*Destexhe et al., 1999*).

## Spatiotemporal dynamics

To study spatiotemporal dynamics in these neural recordings, we adapted our previously introduced method for detecting arbitrarily shaped traveling waves (*Muller et al., 2014*) to multisite ECoG arrays (*Figure 1—figure supplements 2* and *3*). This approach allowed us to characterize and classify the spatiotemporal dynamics during thousands of episodes of spindling activity in many hours of sleep recordings. The method proceeds in three steps: (1) analytic signal representation for characterization of instantaneous signal characteristics at each electrode, (2) center localization at each individual oscillation cycle, and (3) quantification of the spatiotemporal pattern in each oscillation cycle as a function of distance from (or rotation about) the isolated center point. In the following, we describe in detail the method for isolating expanding and rotating waves in multichannel data.

To estimate instantaneous signal characteristics, we employ the well-known analytic signal representation. This approach entails transforming a real-valued timeseries into a complex phasor, whose modulus (length) and argument (angle) in the complex plane represent the signal instantaneous amplitude and phase, respectively. Specifically, if $v_{x,y,t}$ is a real-valued, narrowband timeseries at a point $(x,y), x \in [1, N_c], y \in [1, N_r]$, where $N_c$ and $N_r$ denote the number of rows and columns, and $t \in [1, N_t]$ is the sample number, then its analytic signal representation is

$$V_{x,y,t} = v_{x,y,t} + i\hat{v}_{x,y,t} \tag{1}$$

where $i$ is the complex unit and $\hat{f}$ denotes the Hilbert transform of a signal $f$. The instantaneous phase of $v_{x,y,t}$ is then the argument at each point in this complex sequence

$$\phi_{x,y,t} = Arg(V_{x,y,t}), \tag{2}$$

and instantaneous amplitude is the modulus. At several points in the analysis, results were confirmed with an FIR implementation of the Hilbert transform, in addition to the standard FFT-based approach (*Marple, 1999*). We evaluated phase values at a set of time points $\mathcal{T} = \{t_1, t_2, \dots, t_K\}$ in each spindle

near the positive oscillation peaks. The phase fields were then smoothed using a robust approach (*Garcia, 2010*) for center localization to reduce noise and interpolate values from missing electrodes; note that the smoothed values were not used in the calculations for detecting expanding and rotational wave patterns.

These phase values are then used to capture spatiotemporal dynamics in the multichannel data. To isolate putative expanding or rotating wave centers in each oscillation cycle, we first assess the spatial gradient of phase

$$\vec{g}_{x,y,t_j} \equiv -\nabla\phi_{x,y,t_j} \tag{3}$$

with $t_j \in \mathcal{T}$. For the spatial gradient, derivatives are taken across the two dimensions of space and are approximated by the appropriate forward and centered finite differences. As in previous work, phase derivatives were implemented as multiplications in the complex plane (*Feldman, 2011*; *Muller et al., 2014*).

## Detection of expanding waves

To detect expanding waves, we assess the divergence of the phase gradient field

$$d_{x,y,t_j} = \nabla \cdot \vec{g}_{x,y,t_j}, \tag{4}$$

and define the putative wave source to be that point which satisfies the arg max over space in each cycle

$$\mathcal{S} \equiv (x,y;t_j) = \arg\max_{x,y} \mathrm{d}_{x,y,t_j}, \tag{5}$$

where $\arg\max f(a,b) \equiv \{a,b \,|\, \forall p,q : f(p,q) \leq f(a,b)\}$. This step allows us to find the source for a possible expanding wave in each cycle (step 2, *Figure 1—figure supplement 2*), about which the phase field is then evaluated to quantify the evidence for an expanding wave spatiotemporal organization (step 3, *Figure 1—figure supplement 2*). For this next step, we calculate the circular-linear correlation coefficient $\rho_{\phi,\delta}$ (*Jammalakadaka and Sengupta, 2001*; *Berens, 2009*) between signal phase $\phi$ and radial distance $\delta$ from the source point in the original, *unsmoothed* phase field

$$\rho_{\phi,\delta} = \sqrt{\frac{r_{c\delta}^2 + r_{s\delta}^2 - 2r_{c\delta}r_{s\delta}r_{cs}}{1 - r_{cs}^2}}, \tag{6}$$

where $r_{c\delta}$ represents the Pearson correlation between the cosine of the circular variable $\phi$ and the linear variable $\delta$, $r_{s\delta}$ between the sine of $\phi$ and the variable $\delta$, and $r_{cs}$ between the cosine and sine of $\phi$. This approach allows us to quantify the strength of the spatiotemporal pattern of activity on the array in a single number, which is then compared to the value produced by repeating the calculations many times under random shuffling of the data (blue dotted line, *Figure 1—figure supplement 4A* and Materials and methods – Shuffling Controls).

## Detection of rotating waves

Analogous to the above case, we start by assessing the curl of the phase gradient field

$$\vec{c}_{x,y,t_j} = \nabla \times \vec{g}_{x,y,t_j}, \tag{7}$$

and defining the putative center to be that point which satisfies the arg max over space

$$\mathcal{C} \equiv (x,y;t_j) = \arg\max_{x,y} ||\vec{c}_{x,y,t_j}||. \tag{8}$$

This center point then defines an anchor about which we can pass into a polar coordinate system, describing the distance $\delta$ and rotation angle $\theta$ about that point (step 2, *Figure 1—figure supplement 3*). With the putative rotation center isolated in each oscillation cycle, we then proceed to calculate the circular-circular correlation coefficient $\rho_{\phi,\theta}$ between signal phase $\phi$ and rotation angle $\theta$ (*Fisher, 1993*; *Berens, 2009*) in the original, unsmoothed phase field

$$\rho_{\phi,\theta} = -\frac{\sum_{xy}\sin(\phi_{xy}-\overline{\phi})\sin(\theta_{xy}-\overline{\theta})}{\sqrt{\sum_{xy}\sin^2(\phi_{xy}-\overline{\phi})\sin^2(\theta_{xy}-\overline{\theta})}}, \tag{9}$$

where overbar indicates circular mean

$$\overline{\phi} = Arg\left[\sum_{xy}e^{i\phi_{xy}}\right]. \tag{10}$$

Similar to the previous case, this number $\rho_{\phi,\theta}$ quantifies the evidence for a rotational wave organization on the array, which is then compared to the value derived from a random-shuffling permutation test (red dotted line, *Figure 1—figure supplement 4A* and Materials and methods – Shuffling Controls). For each case, additional control analyses with simulated rotating waves embedded in noise were used to verify the robustness of our approach (*Figure 1—figure supplement 11*). Finally, in the case that both expanding and rotational elements are detected, the pattern is classified as rotational, because sub-patterns of rotational waves tend to be detected as expanding elements (verified in *Figure 1—figure supplement 5*; see also Materials and methods – Average vector field controls).

## Shuffling controls

To quantify the level of spatiotemporal phase flow expected in the data by chance, we implemented a shuffling procedure to establish a permutation-based threshold for both the expanding and rotating wave measures. To do this, we shuffled the phase values in each oscillation cycle randomly across space a number of times (100 or 1000 times in initial tests, then reduced to 25 without changing results), repeating each time the same calculation as for the un-shuffled data. The 99th percentile of the resulting distribution then determines a threshold above which the value for the correlation metric (either for expanding or rotational waves, considered separately) exceeds chance, with the spatial autocorrelation erased.

A possible confound resulting from this shuffling procedure is that the data intrinsically possess some spatial autocorrelation (*Figure 1—figure supplement 1B*), which is ignored by the so-constructed permutation test. To address this point in the context of rotational wave detection, we conceived an additional permutation test control. In this second control, we considered the set of points at a Chebyshev (i.e. King's chessboard) distance $d_i \in [1, d_m]$ from the putative rotation center, where $d_m$ indicates the maximum distance on the electrode array from that point. We then shuffled channels at distances $d_i$ for all $i$, and repeated the calculation for the rotational wave detection. The resulting permuted data have a spatial correlation function identical to that in the un-shuffled data, but with the rotational structure fully destroyed. The 99th percentile cutoff determined from this second control analysis fell within 0.01 of the originally estimated value (3% difference), validating the original shuffling permutation test employed above.

## Simulated data controls

Using simulated expanding waves of the form

$$f(t,\delta) = Ae^{i(\omega t - \kappa\delta)} + \sigma\eta(t), \tag{11}$$

and simulated rotating waves

$$f(t,\theta) = Ae^{i(\omega t - \gamma\theta)} + \sigma\eta(t), \tag{12}$$

where $A$ is the oscillation amplitude, $\omega$ is the oscillation angular frequency, $\kappa$ is the wavenumber, $\gamma$ is the polar wavenumber, and $\eta(t)$ is a Gaussian white noise term, we verified the robustness of our detection approach under noise of varying amplitudes. Note that $\delta$ and $\theta$ are defined with respect to the wave center, left unspecified for simplicity. Oscillation amplitude was set to unity, without loss of generality, and other oscillation parameters were matched to those observed during stage 2 sleep spindles. Oscillation frequency $\omega$ was set to the average instantaneous frequency estimated from 702 spindles in Subject 1 (13.5 Hz), and wavenumbers were adjusted to approximate the wavelengths observed in the data. Varying systematically the level of added noise, we ran the algorithms

for detecting expanding and rotating waves described above for 25 trials at each point and recorded the algorithm's detection performance in each case (mean ± SEM, *Figure 1—figure supplement 11C*). These results illustrate the approximate invariance of our computational approach to random noise.

In another test, we systematically varied the position of the wave center on the simulated 64 electrode array, for both expanding and rotational waves (*Figure 1—figure supplement 11D*). Parameters were set as above, and simulations were again run over 25 trials at each point. This test probed the sensitivity of the rotational detection approach to border effects, which is expected to be negligible at the encountered noise levels in comparison to the thresholds established by the permutation controls (dotted lines, *Figure 1—figure supplement 11D*).

In a third test, we verified that the spatiotemporal patterns observed here are not due to variations in spindle frequency, which are known to occur along the rostro-caudal axis (*Peter-Derex et al., 2012*). To do this, we re-ran our analysis on one stage 2 sleep session containing 179 spindles in Subject 1, generating surrogate data as follows. Each electrode evolved in time according to its mean instantaneous frequency during the spindle, but with a randomized initial phase angle. These surrogate data thus possessed the same frequency content on average as in the original data, but with their spatial organization of phase removed. In this control, both rotating and expanding wave patterns were highly decreased (3.5% and 0.8% of cycles, respectively, compared to 64% and 14% in the original data).

## Average vector field controls

The algorithmic classification of wave patterns in individual oscillation cycles involves several steps, and we wanted to make an independent check to verify these results. To do this, we adopted a re-centered averaging approach, shifting the vector field of propagation directions from the smoothed phase fields at each oscillation cycle to the putative rotation center (red dots, *Figure 1—figure supplement 5*), and taking the circular mean (*Fisher, 1993*; *Berens, 2009*) of propagation direction at each point. Performing the calculation in this way prevents regions with noise or high phase gradient magnitude from dominating the result. The obtained vector fields for rotational TPF (*Figure 1—figure supplement 5A*) and expanding (*Figure 1—figure supplement 5B*) waves illustrate the accuracy of the algorithm and the general validity of our classification approach in separating waves into expanding and rotational groups.

## Phase map correlation analysis

To quantify the precision of repeated spatiotemporal patterns during across spindle oscillations over several minutes of data, we calculated the circular-circular correlation between phase values in individual oscillation cycles. For two phase maps $\alpha_{x,y}$ and $\beta_{x,y}$, the circular correlation is defined as above (*Fisher, 1993*; *Berens, 2009*)

$$\rho_{\alpha,\beta} = \frac{\sum_{xy}\sin(\alpha_{xy}-\overline{\alpha})\sin(\beta_{xy}-\overline{\beta})}{\sqrt{\sum_{xy}\sin^2(\alpha_{xy}-\overline{\alpha})\sin^2(\beta_{xy}-\overline{\beta})}}\,. \tag{13}$$

This correlation value defined between individual phase maps then constitutes elements of an $M \times M$ matrix $C_{ij}$, where $M$ is the number of isolated oscillation cycles in question:

$$C_{ij} = \begin{bmatrix} 1 & \rho_{1,2} & \rho_{1,3} & \rho_{1,4} & \cdots & \rho_{1,M} \\ \rho_{2,1} & 1 & \rho_{2,3} & \rho_{2,4} & \cdots & \rho_{2,M} \\ \rho_{3,1} & \rho_{3,2} & 1 & \rho_{3,4} & \cdots & \rho_{3,M} \\ \rho_{4,1} & \rho_{4,2} & \rho_{4,3} & 1 & \cdots & \rho_{4,M} \\ \vdots & \vdots & \vdots & \vdots & \ddots & \vdots \\ \rho_{M,1} & \rho_{M,2} & \rho_{M,3} & \rho_{M,4} & \cdots & 1 \end{bmatrix}$$

where all elements in the diagonal and lower triangle ($C_{ij} \forall i \ge j$) are not considered, without loss of generality. We then construct this matrix for all cycles in an individual wave classification (all cycles, expanding, rotational), and consider the cumulative distribution function (CDF) of values in the upper triangle (*Figure 3A*). To construct the CDF in the permutation case (*Figure 3A*, purple

line), we first randomly shuffled the phase maps in all spindle cycles and then proceeded with the calculation as normal.

## Code availability

A MATLAB toolbox for analysis of traveling waves and complex spatiotemporal dynamics in noisy multisite data is available as an open-source release on BitBucket: http://bitbucket.org/lylemuller/wave-matlab

## Acknowledgements

The authors would like to thank the clinical patients for their participation in the research, in addition to T Bartol, F Chavane, A Destexhe, and CF Stevens for helpful discussions. The authors are grateful to the Swartz Foundation, the Howard Hughes Medical Institute, the Office of Naval Research, and NIH for support.

## Additional information

### Funding

| Funder | Grant reference number | Author |
| --- | --- | --- |
| Swartz Foundation | | Terrence J Sejnowski |
| Howard Hughes Medical Institute | | Terrence J Sejnowski |
| Office of Naval Research | MURI N000141310672 | Sydney S Cash<br>Eric Halgren<br>Terrence J Sejnowski |
| National Institutes of Health | R01EB009282 | Terrence J Sejnowski |
| National Institutes of Health | 5T32EY20503-5 | Lyle Muller |
| Bial Foundation | BIAL 220/12 | Giovanni Piantoni |
| Office of Naval Research | N000141210299 | Terrence J Sejnowski |

The funders had no role in study design, data collection and interpretation, or the decision to submit the work for publication.

### Author contributions

LM, TJS, Conception and design, Analysis and interpretation of data, Drafting or revising the article; GP, SSC, EH, Acquisition of data, Analysis and interpretation of data, Drafting or revising the article; DK, Analysis and interpretation of data, Drafting or revising the article

### Author ORCIDs

Lyle Muller, http://orcid.org/0000-0001-5165-9890
Giovanni Piantoni, http://orcid.org/0000-0002-5308-926X
Terrence J Sejnowski, http://orcid.org/0000-0002-0622-7391

### Ethics

Human subjects: Patients with longstanding pharmacologically resistant complex seizures gave fully informed consent according to NIH guidelines as monitored by the local Institutional Review Board (Massachusetts General Hospital). Electrocorticogram (ECoG) recordings during natural sleep were made over the course of clinical monitoring for spontaneous seizures. Electrode placement was determined solely by clinical criteria, with electrode grids usually spanning the Sylvian fissure and multiple lobes of the cerebral cortex (frontal, parietal, and temporal). Patients were informed that participation in the research would not alter their clinical treatment in any way, and that they may withdraw their consent at any time without jeopardizing clinical care (see Materials and Methods - Subjects).

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
