## [Decision Letter]

Thank you for submitting your article "Repeating circular waves enable strengthening of large-scale neural assemblies during sleep spindles in human cortex" for consideration by *eLife*. Your article has been favorably evaluated by Timothy Behrens (Senior Editor) and three reviewers, one of whom is a member of our Board of Reviewing Editors.

The reviewers have discussed the reviews with one another and the Reviewing Editor has drafted this decision to help you prepare a revised submission.

This work is an analysis of spindle oscillations recorded from human subdural grids placed for the purpose of localizing epileptic seizures. The authors examine the spatio-temporal structures of spindles during sleep in human, and their possible role in memory consolidation. The techniques used are developed from their earlier work where waves were found in monkey visual cortex. Used here in considering both amplitude and phase information, the authors show that spindles exhibit a sophisticated spatio-temporal structure, with the majority of spindles traveling from temporal to parietal to frontal (TPF) cortex. The authors suggest that such traveling waves could serve to facilitate spike-timing dependent plasticity between anatomically remote areas, which would otherwise be difficult due to long axonal delays. The analysis is rigorous and the application of methods to detect propagating waves to spindles is novel. Spindles are of high interest for memory consolidation, and this dataset is highly valuable.

The revision requirements are three-fold.

1) An expansion and clarification of methodological aspects is needed in various places. Specific points are as follows.

a) Rotating spindle waves: The presented data and analyses on spindle oscillations are clear. They rely on a large dataset of spindle oscillations and the provided movies give a good sense of the observations made. However, clarifications are needed. First, the authors should incorporate a definition of what they consider as rotating and expanding waves, as well as the specific criterion used to distinguish these. Definitions should appear in the main text, and the supplementary material should include the specific criteria used for the classification of spatio-temporal patterns of activity. In the absence of this, it remains quite difficult to validate the method.

The methods present clearly the identification of the center point of the oscillation, important to perform a polar parameterization, thus essential to extract rotating or expanding waves. This step is well described and the method relies on the computation of gradients and curl. It would be important to know:

i) If the authors pre-processed the data and if smoothing occurred to compute these differential elements. In particular, filtering is indicated in Figure 1—figure supplement 8. Was a similar filtering used in the identification of waves? If this was not the case, how did the authors correct high frequency signals that may be amplified in the computation of differential quantities?

ii) Once the polar coordinate system has been derived and the phase map computed, how did the classification take place and what were the error rates accepted for this classification? The statement about the proportion of spindle oscillations classified as rotating waves of ~50% remains a little bit hard to appreciate since there are only 2 classes (plus "complex" waves that are those not satisfying their criteria of rotating or expanding wave). Moreover, the analysis of Figure 1—figure supplement 7 provides a more contrasted view of this result: rotational waves are not necessarily the leading form of oscillation, except for subject 1 (for other subjects, spindles were generally classified as complex).

iii) The authors indicate that among the rotating waves, they identified a clear bias towards TPF direction of rotation. Was this observation supported by a statistical test?

iv) The waves observed indeed reflect a peak between 3-5 m/s, but also a very high variance. From the individual curves, it seems that we see the superposition of two speed distributions centered at distinct values. Could the authors comment on the variations of the speed value provided?

b) The analysis of intra-class similarity is nicely developed. However, for the level of synchronization, it was unclear how the millisecond precision was computed. Particularly, it was unclear what was meant by "in two cycles with the highest similarity". Could the authors elaborate in the Methods or in the text on this choice?

c) A rigorous demonstration that spindles are waves would be a highly novel finding. However, given the limited size of the dataset, and the relationship of spindles to the underlying anatomy being unclear, additional analyses and explanations are needed.

i) Discrepancy with "local spindles", different frequencies of controparietal/frontal spindles finding, and relationship to underlying anatomy. Much previous work concludes that human spindles are "local" and that spindle frequencies vary between areas (i.e. Andrillon et al. 2011, Peter-Derex et al. 2012, Nir et al. 2014). However, here the statement is made that this is in fact not so and the explanation is provided that this might be due to using a threshold for detection. It is unclear how this analysis supports this statement. What would the wave-detection approach show if these spindles would in fact be local? All that is done here is to fit a field of vectors in 2D space and then to see if a curl can be fit. No goodness of fit is assessed (only that it is significant). But presumably this would also work for fields where only subsets of the vectors are fit, with the others random. Also, spindles have different frequencies depending on anatomical origin as well as in time relative to spindle onset. Could these frequency changes "generate" the appearance of waves (or vice-versa)?

ii) Structure of waves. It was unclear whether the origin (center) of waves remained constant, changed wave-by-wave or changed slowly. Figures are shown "re-centered", so it isn't apparent where the actual center was. Also, how sensitive is this method to edge effects, i.e., is it equally sensitive to detect centers everywhere on the array or only in the center? Overall, the location of the centers and the underlying anatomy should be clarified. The authors suggest a TPF direction of spread, so does this indicate center preferentially in temporal lobe?

2) A toning down of interpretation and claims and situating the work more appropriately. That is, STDP and memory consolidation aspects aren't actually measured. Specifically:

a) The authors should consider adjusting their title somewhat in toning down the claims (i.e., "… enable strengthening of large-scale neural assemblies…") to more appropriately reflect the main aspects of the paper.

b) Mechanistic interpretation is not supported. The Abstract talks extensively about synapses and STDP, neither of which is measured here. This should be clarified (e.g., interesting suggestions, but cannot be assessed using this data). Same for "spiking activity" – all that is measured is γ band power. Statements such as "groups of spikes" are not supported by this data. Others have measured spindles and spikes simultaneously in humans (i.e. Andrillon et al., as cited), showing a complex relationship.

c) Another suggestion is that the authors expand on their Figure 2 description/illustration. For example, could the authors say something about spindle mechanisms (RT cells and mutual inhibition etc.), and how such spikings illustrated in Figure 2 translate to EPSPs (IPSPs?). Haider et al. is cited to justify LFP and synaptic current tight couplings, but this includes IPSPs. Related to this is the consideration of different STDP rules for different cell types (e.g., see Abbott and Nelson, Nature Neuroscience 2000).

In essence, since much has been published regarding STDP, spindling mechanisms, memory consolidation etc., it would be good for the authors to put these various pieces together in their suggestions for the general reader to appreciate more fully, and so that interpretations/limitations/assumptions are clear in what is being said. As it stands, it comes across as overly simplistic regarding the timing relationships and LTP/LTD (Figure 2) and traveling waves with STDP aspects and so on.

3) Clearly describing/demonstrating that a differentiation between the two types of 'waves' is possible along with what is the current consensus in the literature – 'local spindles' and how it was ruled out with their analyses. This is unclear at present. Perhaps some of the unclassified waves are of that nature? That is statistics and classification are not always clear [see point (1) above].

Additional points to address:

i) In general, please provide more statistical support and more information on the statistical tests and the data analysis procedures used. It is stated that the code is available on bitbucket, but when looked for by one of the reviewers, access was not possible, thus preventing a better understanding of the types of tests used.

ii) How were the recordings referenced? Phase analysis is highly sensitive to this.

iii) What Figure 4 quantifies is unclear – please clarify how this is calculated. Can one show the same data for expanding waves and show this critical analysis for multiple patients?

---

## [Author Response]

*[…] The revision requirements are three-fold.*

*1) An expansion and clarification of methodological aspects is needed in various places. Specific points are as follows.*

*a) Rotating spindle waves: The presented data and analyses on spindle oscillations are clear. They rely on a large dataset of spindle oscillations and the provided movies give a good sense of the observations made. However, clarifications are needed. First, the authors should incorporate a definition of what they consider as rotating and expanding waves, as well as the specific criterion used to distinguish these. Definitions should appear in the main text, and the supplementary material should include the specific criteria used for the classification of spatio-temporal patterns of activity. In the absence of this, it remains quite difficult to validate the method.*

This is an excellent suggestion; we have now inserted explicit definitions for rotating and expanding waves into the main text (second paragraph). We have further collected the specific classification criteria into two discrete Methods subsections (“Detection of expanding waves” and “Detection of rotating waves”), in addition to adding another figure supplement detailing the process for detecting expanding waves (Figure 1—figure supplement 2). We agree that these additions greatly improve the presentation of analysis techniques.

*The methods present clearly the identification of the center point of the oscillation, important to perform a polar parameterization, thus essential to extract rotating or expanding waves. This step is well described and the method relies on the computation of gradients and curl. It would be important to know:*

*i) If the authors pre-processed the data and if smoothing occurred to compute these differential elements. In particular, filtering is indicated in Figure 1—figure supplement 8. Was a similar filtering used in the identification of waves? If this was not the case, how did the authors correct high frequency signals that may be amplified in the computation of differential quantities?*

As in our previous work (Muller et al., 2014), the robust nonparametric smoothing technique of Garcia (2010) is employed before center localization (and, here, computation of differential quantities). It is important to note that after center localization, however, the phase correlation quantities are calculated on the *unsmoothed* phase fields, in order to prevent any smoothing effects from contaminating wave detection results. This was stated in the original manuscript, and in the revised submission we have striven further to clarify this important point (subsection “Spatiotemporal dynamics”, second paragraph and subsection “Detection of expanding waves”, first paragraph).

*ii) Once the polar coordinate system has been derived and the phase map computed, how did the classification take place and what were the error rates accepted for this classification?*

In order to validate our computational approach for the 48-96 channel ECoG arrays, we analyzed the performance of our algorithm in simulated data under varying degrees of noise. Because our method takes a statistical approach to wave detection, the algorithm detects waves with fidelity up to high noise levels, as opposed to algebraic methods that tend to break down in the presence of noise (Wong and Yip, Pattern Recognition42, 2009). While we did not include these verification tests in the original manuscript, this comment has emphasized to us the importance of doing so, and we now include these tests in the revised submission (see Figure 1—figure supplement 11 and Figure 1—figure supplement 12). As demonstrated in Figure 1—figure supplement 11, our approach robustly identifies simulated expanding and rotational waves while reporting no detections with only noise. We thank the Reviewers for this helpful comment.

*The statement about the proportion of spindle oscillations classified as rotating waves of ~50% remains a little bit hard to appreciate since there are only 2 classes (plus "complex" waves that are those not satisfying their criteria of rotating or expanding wave). Moreover, the analysis of Figure 1—figure supplement 7 provides a more contrasted view of this result: rotational waves are not necessarily the leading form of oscillation, except for subject 1 (for other subjects, spindles were generally classified as complex).*

This was unclear in the original submission, and we thank the Reviewers for making this important point. The critical ambiguity here was to designate cycles that are not classified as rotational or expanding waves as *complex*, rather than simply *unclassified*. Our initial logic was that, because we observe very few spindle oscillation cycles that are fully synchronized across the array (i.e. temporal extent less than 10 ms), we termed the unclassified oscillation cycles “complex”; however, this does not allow for the fact that some spindle cycles may in fact be “local”. We have thus updated the term “complex” in the manuscript simply to “unclassified”, and explained this more clearly in the revised text (main text, seventh paragraph and Figure 1—figure supplement 9). We thank the Reviewers for this helpful suggestion.

*iii) The authors indicate that among the rotating waves, they identified a clear bias towards TPF direction of rotation. Was this observation supported by a statistical test?*

We have taken this comment into account and now provide the results of an exact binomial test on the bias towards TPF direction (main text, second paragraph). The bias is significant for each individual subject (p < 10^-3^ in each case), as well as for the full population (p < 10^-10^, 14,796 TPF cycles, 21,272 total). We thank the Reviewers for this helpful suggestion.

*iv) The waves observed indeed reflect a peak between 3-5 m/s, but also a very high variance. From the individual curves, it seems that we see the superposition of two speed distributions centered at distinct values. Could the authors comment on the variations of the speed value provided?*

These speed distributions are similar in extent to those observed in previous literature (Rubino et al., Nature Neuroscience9, 2006, cf. their Supplementary Figure 2), and are strongly peaked near values corresponding to physiological measurement. Looking further into the distributions, high variance is not apparent: from the 20th to the 80th percentiles, speeds in the full distribution vary over a tight range (3-9 m/s), corresponding approximately to 20-50 ms of travel time across the array. Speed distributions for individual subjects are similar (from 20-80th percentile: 4-10, 3-8, 3-10, 2-3, and 4-13 m/s, respectively). The variation in these distributions is due in part to measurement noise, with noisy but statistically significant events showing up as high-speed waves on the array. We thank the reviewer for this helpful comment, which we have now pointed out in the main text (second paragraph).

Further, true variations in wave speed can certainly result in better or worse alignment of spikes across distant cortical regions, as discussed in our Figure 2. As mentioned in our response to point 2c, we obtained in previous work a mathematical expression for the expected weight change by STDP in this case, which allows us to understand the effect of speed variability on the plasticity mechanisms discussed here. We fully agree to include discussion of this important point in the main text (last paragraph), and we thank the reviewers for this helpful comment.

*b) The analysis of intra-class similarity is nicely developed. However, for the level of synchronization, it was unclear how the millisecond precision was computed. Particularly, it was unclear what was meant by "in two cycles with the highest similarity". Could the authors elaborate in the Methods or in the text on this choice?*

We agree that it is useful for us to clarify this important point. In the revised submission, we have updated the main text to describe this calculation more clearly (fifth paragraph).

*c) A rigorous demonstration that spindles are waves would be a highly novel finding. However, given the limited size of the dataset, and the relationship of spindles to the underlying anatomy being unclear, additional analyses and explanations are needed.*

We appreciate this positive comment on our work, and we agree that reporting spindles are spatiotemporally organized into global traveling waves is an important result with strong implications for spindles’ established function in sleep-dependent consolidation of long-term memory. In the revised submission, we have striven to clarify analysis procedures and to provide additional evidence to further support these results (detailed point-by-point in the responses below).

*i) Discrepancy with "local spindles", different frequencies of controparietal/frontal spindles finding, and relationship to underlying anatomy. Much previous work concludes that human spindles are "local" and that spindle frequencies vary between areas (i.e. Andrillon et al. 2011, Peter-Derex et al. 2012, Nir et al. 2014). However, here the statement is made that this is in fact not so and the explanation is provided that this might be due to using a threshold for detection.*

In accordance with point 1.ii above, we recognize the importance of clarifying the relationship of the observed waves to previous observations of “local spindles”, and we thank the Reviewers for pointing this out. In the revised submission, we now denote oscillation cycles unclassified by the algorithm simply as “unclassified”, rather than “complex” (Figure 1—figure supplement 9). This shift in terminology now allows for the possibility of local spindles, which constitute approximately half of the observed events (< 40% channel involvement) in previous work (Andrillon et al., 2011). We acknowledge the possibility that it is in fact the set of “global” spindles which are robustly (and almost completely) organized as rotational and expanding traveling waves across the cortical surface (main text, seventh paragraph).

In fact, this variation in spatial extent suggests an interesting functional role: while in some cases, memory content will be specialized and thus confined to a single cortical region (Khazipov et al., Nature432, 2004), in other cases memories will contain content across modalities (visual, auditory, or cognitive) that must be distributed across regions, an organization which requires the spatiotemporal patterns observed here (as illustrated in Figure 2). We have now updated our submission to include this point (main text, last paragraph), and we thank the Reviewers for their helpful input.

Finally, it is important to emphasize, as in our original submission, that our phase-based method represents a well-controlled, robust approach to detecting waves in noisy multichannel data. As such, it is not surprising that we are able to detect broad spatiotemporal patterns of activity in cortex where past studies have detected only noisy patches of spindle activity in a handful of electrodes with amplitude-duration detection (Mölle et al., J Neurosci22, 2002). Our approach is robust to false wave detections and sensitive to coherent activity patterns embedded in noise. In order to clarify these points, we have included several control analyses demonstrating the validity of our approach. We detail these specifically in the response to the next few points, and we have updated the manuscript to clarify these aspects of our method.

*It is unclear how this analysis supports this statement. What would the wave-detection approach show if these spindles would in fact be local? All that is done here is to fit a field of vectors in 2D space and then to see if a curl can be fit. No goodness of fit is assessed (only that it is significant).*

We understand and appreciate the reviewers’ concern on this point. We must emphasize, however, that the above characterization of our method is inaccurate. As stated in the Methods, we only use the curl operation to estimate the putative center of rotation (step 2, Figure 1—figure supplement 3), after which we calculate the *circular-circular correlation* between signal phase and rotation about this point (step 3, Figure 1—figure supplement 3), which “quantifies the strength of the spatiotemporal pattern of activity … in a single number” (subsection “Detection of expanding waves”). This number, ϱ_φ,θ_, represents in itself a “goodness of fit”, whose absolute value ranges from 0 (no pattern) to 1 (fully organized rotational pattern).

In addition to this, we employed the average vector field quantification as an independent control (Figure 1—figure supplement 5). In both the expanding and rotating wave cases, this control demonstrates that we correctly detect and separate the two classes of waves considered, providing an additional demonstration of “goodness of fit”, calculated in a different but complementary manner to our primary wave detection approach.

Further, it is important to note that the expanding and rotating patterns extend out to the edges of the array in Figure 1—figure supplement 5, indicating in accordance with the phase correlation measure that these waves are truly global coherent activity patterns. In the revised manuscript, we have added a figure supplement illustrating the full distribution of propagation directions at each point in the average vector field for rotational TPF waves in all subjects (gray lines, Figure 1—figure supplement 6), in addition to the circular mean (black arrows, as in other figures), in order to clearly demonstrate that the observed patterns are indeed widespread and global.

We hope that these additional explanations fully address the reviewers’ concern on this point. In accordance with this, we have striven in our revision to further clarify our computational algorithm for wave detection and the controls validating this approach.

*But presumably this would also work for fields where only subsets of the vectors are fit, with the others random.*

In order to address this concern directly, we have included an additional control comparing detection of a global rotating wave versus a very local one (6 electrodes, or 9.4% of the 64 channel electrode array; Figure 1—figure supplement 12). This control demonstrates that if few electrodes participate in the phase pattern, as would be the case with a fully “local” spindle, this will not be detected. This control illustrates that finding high ϱ_φ,θ_ is strongly indicative of a distributed and global phase pattern. This result is further supported by the average vector field control calculations (Figure 1 and Figure 1—figure supplement 5–Figure 1—figure supplement 7, as detailed in the point above).

Further, it is important to note that this approach, together with the sampling density available in this study, emphasizes global phase patterns and will be less sensitive to smaller, intermediate patterns. This is perhaps why we see a robust rotating TPF wave organization in the vector field average over all cycles in each individual subject (Figure 1—figure supplement 7). Thus, while we can be confident that the rotating wave organization reported appears with *at least* the occurrence rate reported here, the true incidence rate is most likely higher (including any intermediate waves). This point will be pursued in future work, utilizing higher density electrode arrays.

We thank the reviewers for this interesting and helpful comment, and we believe that it has greatly improved our work.

Also, spindles have different frequencies depending on anatomical origin as well as in time relative to spindle onset. Could these frequency changes "generate" the appearance of waves (or vice-versa)?

This control was one of the first concerns raised in validating our results. In order to test this, we re-ran our analysis on one stage 2 sleep session from Subject 1 with surrogate data, where each electrode evolved in time with its mean instantaneous frequency during the spindle, with a randomized initial phase angle. In this control, both rotating and expanding wave patterns were highly decreased (3.5% and 0.8% of cycles with rotating and expanding waves, respectively, in the control simulation versus 64% and 14% in the original). This control demonstrates that frequency shifts cannot explain our results, indicating that the observed waves are due to a true phase shift across channels. To create these phase shifts, active corticocortical, thalamocortical, and corticothalamic connections are necessary. We plan to address detailed mechanisms for such network-level generation of the spatiotemporal activity patterns in future modeling work. We thank the reviewer for addressing this important point, the control for which we now discuss in the Methods (subsection “Simulated data controls”, last paragraph).

*ii) Structure of waves. It was unclear whether the origin (center) of waves remained constant, changed wave-by-wave or changed slowly. Figures are shown "re-centered", so it isn't apparent where the actual center was. Overall, the location of the centers and the underlying anatomy should be clarified. The authors suggest a TPF direction of spread, so does this indicate center preferentially in temporal lobe?*

We agree that this is an important point, and welcome the opportunity to expand upon it. In the revised submission, we have included an additional figure (Figure 1—figure supplement 8) to illustrate the distribution of rotation center (across all 10,944 rotating TPF waves in Subject 1) and movement between cycles (aggregated across TPF waves in all subjects). These additional panels demonstrate that the center of rotation distribution appears to be concentrated just dorsal to the temporal lobe, aligned on the Sylvian fissure (Figure 1—figure supplement 8). The rotation center further appears to move slowly across the surface of the cortex, if at all (Figure 1—figure supplement 8), in agreement with the impression from Video 1. We believe that this analysis, along with our response to the following query, fully addresses the reviewers’ concerns on this point.

Also, how sensitive is this method to edge effects, i.e., is it equally sensitive to detect centers everywhere on the array or only in the center?

We have tested this point directly using simulated rotating waves in surrogate data, and in the revised submission, we have included this analysis in an additional figure (Figure 1—figure supplement 11). In short, we are able to detect noisy waves embedded across the array. Though sensitivity is decreased at the edges, we are confident that the distribution of detected rotation centers is unaffected by issues of sensitivity.

*2) A toning down of interpretation and claims and situating the work more appropriately. That is, STDP and memory consolidation aspects aren't actually measured. Specifically:*

*a) The authors should consider adjusting their title somewhat in toning down the claims (i.e., "… enable strengthening of large-scale neural assemblies…") to more appropriately reflect the main aspects of the paper.*

While we do believe our manuscript contains intriguing preliminary evidence towards strengthening of repeating patterns, and similar evidence in visual cortex of the mouse has previously been interpreted as involving plasticity-dependent mechanisms (Han et al., Neuron60, 2008), we fully understand the reviewers’ concern on this point and agree that adjusting the title can more accurately reflect the analyses presented in the paper. In line with this suggestion, we have updated the title in our revised submission to:

“Rotating waves during human sleep spindles organize global patterns of activity that repeat precisely through the night”.

We thank the reviewers for this helpful comment.

*b) Mechanistic interpretation is not supported. The Abstract talks extensively about synapses and STDP, neither of which is measured here. This should be clarified (e.g., interesting suggestions, but cannot be assessed using this data). Same for "spiking activity" – all that is measured is γ band power. Statements such as "groups of spikes" are not supported by this data. Others have measured spindles and spikes simultaneously in humans (i.e. Andrillon et al., as cited), showing a complex relationship.*

We understand this point, and we regret that our initial submission did not clearly review the background knowledge from animal electrophysiology on this point. A wealth of evidence from intracellular and single-unit recordings presented over two decades shows that during spindles, cells in neocortex receive strong excitatory input from thalamus and concentrate spikes at the peak of depth negativity (Contreras and Steriade, J Neurosci15, 1995; Contreras and Steriade, J Physiol 490, 1996; Kandel and Buzsaki, J Neurosci 17, 1997; Peyrache et al., PNAS 108, 2011). In this work, we analyze high-γ power (HGP), which is a well-accepted proxy for spiking activity in the field (Ray et al., J Neurosci 28, 2008; Manning et al., J Neurosci 29, 2009; Ray and Maunsell, PLoS Biology 9, 2011; Ray, Curr Opin Neurobio 31, 2015). Our results are fully in line both with those obtained in animal recordings and previous work in human – specifically Andrillon et al. (2011), whose Figure 4 matches very closely with Figure 1 in Peyrache et al. (2011).

Note that phase 0 in Andrillon et al. (2011) corresponds to π in Peyrache et al. (2011), since Peyrache et al. took deep layer LFP as reference. We believe that the preponderance of evidence, both from previous animal electrophysiology and from human recordings, is so strong on this point that our demonstration of preferred phase for HGP activity (Figure 4) should be viewed as mere confirmation of previous work.

At the same time, we do fully understand that neither spikes nor STDP were directly measured in this study, only proxies for each; thus, while we took care not to misrepresent the results we provide as definitive evidence for this effect, we have revised the manuscript accordingly, ensuring that all functional statements are carefully supported by the data analyses presented here and knowledge from previous studies in animal electrophysiology. Finally, we have updated the main text to further clarify our use of high-γ power, including a clarification of terminology (“HGP” versus “broadband power shift”).

*c) Another suggestion is that the authors expand on their Figure 2 description/illustration. For example, could the authors say something about spindle mechanisms (RT cells and mutual inhibition etc.), and how such spikings illustrated in Figure 2 translate to EPSPs (IPSPs?). Haider et al. is cited to justify LFP and synaptic current tight couplings, but this includes IPSPs.*

Related to the previous point, we have striven in the revised manuscript to relate the known mechanisms of cortical spindling activity to our results. Specifically, it is known that neocortical networks receive strong rhythmic excitatory volleys from thalamocortical cells, driving the spindle rhythm (see citations above). These strong excitatory inputs modulate firing of cortical pyramidal cells and interneurons (Peyrache et al., 2011), whose EPSPs and IPSPs, respectively, will contribute to the cortical LFP during the spindles measured here.

In order to elaborate on the spiking activity illustrated in Figure 2, we have more clearly emphasized in the revised text that the spikes considered originate from pyramidal cells, the dominant contributor to the white matter tracts (Sholl, The Organization of the Cerebral Cortex, 1956; Schüz and Braitenberg, in Cortical Areas: Unity and Diversity, 2003). It is these excitatory long-range axons that serve to link distant cell groups across the cortex, forming the basis for distributed long-term memories. It is thus their spiking activity – and resulting EPSPs – on which we focus. In accordance with this point, we have clarified the discussion of Figure 2 in the main text (first, fourth and fifth paragraphs).

*Related to this is the consideration of different STDP rules for different cell types (e.g., see Abbott and Nelson, Nature Neuroscience 2000).*

*In essence, since much has been published regarding STDP, spindling mechanisms, memory consolidation etc., it would be good for the authors to put these various pieces together in their suggestions for the general reader to appreciate more fully, and so that interpretations/limitations/assumptions are clear in what is being said. As it stands, it comes across as overly simplistic regarding the timing relationships and LTP/LTD (Figure 2) and traveling waves with STDP aspects and so on.*

In context of the previous point, we are most interested in the spiking (and plasticity) behavior of neocortical pyramidal cells, making long-range projections to pyramidal cells and interneurons in cortex. In previous work, we have studied this problem theoretically and provided an exact analytical result (Muller et al., Frontiers in Computational Neuroscience5, 2011). Further work has generalized this to different plasticity rules (Luz and Shamir, PLoS Computational Biology12, 2016), such as that observed in excitatory-to-inhibitory connections (cf. their Equation 4). Thus, while we present here a simplified schema to illustrate the functional consequences of the observed spatiotemporal activity patterns, we do also understand the system from a theoretical and mathematical perspective. We thank the reviewer for pointing out this initial oversight, and we have updated the main text to mention previous theoretical work and its connection to our submission (last paragraph). Finally, we note that this theoretical understanding also allows us to understand the functional implications for the variance in distribution of wave speed (cf. point 1.iv).

*3) Clearly describing/demonstrating that a differentiation between the two types of 'waves' is possible along with what is the current consensus in the literature – 'local spindles' and how it was ruled out with their analyses. This is unclear at present. Perhaps some of the unclassified waves are of that nature?*

In accordance with point 1.ii above, we appreciate this point and have updated the revised manuscript to include precisely the possibility that some of the unclassified waves are local in nature (main text, seventh paragraph), allowing us to more carefully contextualize our work in relation to previous results. We have also clarified our explanation of the definitions and method for distinguishing rotating from expanding waves, in addition to emphasizing the validation of this distinction in Figure 1—figure supplement 5.

*That is statistics and classification are not always clear [see point (1) above].*

In accordance with point (1) above, we have carefully considered the presentation of methods and statistical tests, and in our revised submission, we have expanded our explanations and the details provided for all calculations and statistical tests.

Additional points to address:

*i) In general, please provide more statistical support and more information on the statistical tests and the data analysis procedures used. It is stated that the code is available on bitbucket, but when looked for by one of the reviewers, access was not possible, thus preventing a better understanding of the types of tests used.*

We welcome the opportunity to expand on the Methods employed in this work. In line with this and other reviewers’ comments, we have moved definitions for analysis terms to the main text, included more information on statistical tests, and expanded the description of our algorithmic approach.

*ii) How were the recordings referenced? Phase analysis is highly sensitive to this.*

In the Methods section of the original manuscript, we stated that “One strip of electrodes positioned over the pial surface and facing the skull served as the reference during the recordings; results were additionally verified using an average reference”. In the revised manuscript, we have clarified the reference for the recordings and that we replicated our results with an average reference, in order to rule out any sensitivity in our analysis to reference effects. At the same time, however, we note that while our results are unaffected by such re-referencing techniques, this is not the preferred approach when analyzing phase in electrode recordings (Shirhatti et al., Neural Computation 28, 2016).

iii) What Figure 4 quantifies is unclear – please clarify how this is calculated. Can one show the same data for expanding waves and show this critical analysis for multiple patients?

We are certainly interested in this analysis and welcome the opportunity to expand on this point. In the revised manuscript, we have clarified the presentation of Figure 4 and included a figure supplement illustrating the results for expanding waves (Figure 4—figure supplement 1). In addition, we have now included an additional analysis showing this result for multiple subjects (Figure 4—figure supplement 2). We believe these additional analyses will address the point raised by the Reviewers, and thank them for raising this important point.